# Transient Phase Clusters in a Two-Population Network of Kuramoto Oscillators with Heterogeneous Adaptive Interaction

**DOI:** 10.3390/e25060913

**Published:** 2023-06-09

**Authors:** Dmitry V. Kasatkin, Vladimir I. Nekorkin

**Affiliations:** A.V. Gaponov-Grekhov Institute of Applied Physics of the Russian Academy of Sciences, 46 Ul’yanov Str., 603950 Nizhny Novgorod, Russia

**Keywords:** phase oscillators, adaptive couplings, heterogeneous interactions, synchronization, transient cluster states, adaptive networks

## Abstract

Adaptive interactions are an important property of many real-word network systems. A feature of such networks is the change in their connectivity depending on the current states of the interacting elements. In this work, we study the question of how the heterogeneous character of adaptive couplings influences the emergence of new scenarios in the collective behavior of networks. Within the framework of a two-population network of coupled phase oscillators, we analyze the role of various factors of heterogeneous interaction, such as the rules of coupling adaptation and the rate of their change in the formation of various types of coherent behavior of the network. We show that various schemes of heterogeneous adaptation lead to the formation of transient phase clusters of various types.

## 1. Introduction

Complex networks consisting of a large number of interconnected elements play an important role in society, nature and technology [1,2,3]. In a wide class of network systems, adaptive networks are attracting increasing attention. A feature of such networks is the change in the strength of connections between nodes depending on their current states. In general, the dynamics of adaptive networks can be represented as a two-level process, consisting of the coevolution of the states of network elements and interelement connections [4,5]. Adaptive network models appear in the description of processes in a wide range of applications, including physics, chemistry, biology, neuroscience, sociology and other fields (see reviews [6,7] and references therein).

One of the approaches to studying the dynamics of adaptive networks is associated with the concept of a phase description developed in the works of Kuramoto [8,9]. This approach has demonstrated its effectiveness in describing and studying various manifestations of synchronization processes in complex networks with interelement connections that are unchanged in time [10,11,12]. In the past few years, models of adaptively coupled phase oscillators have been extensively studied [13,14,15,16,17,18,19,20,21,22,23,24,25,26,27]. As a rule, the networks contain a different structure of connections, including all-to-all, random and ring topology, assuming the identity of the adaptation rules of the couplings. Such adaptive networks exhibit different modes of collective behavior such as multicluster synchronization modes with hierarchical organization [22]; partial synchronization modes, including solitary [27] and chimera [22] states; appearance of modular structure [17,18,19]; transient sequential dynamics [4]; and mixed dynamics [28,29,30].

An important feature that can have a fundamental impact on the collective behavior of a network is the heterogeneity of adaptive couplings. Such feature of interactions naturally arises in neuronal systems when neurons in populations from different areas of the brain interact through synaptic connections characterized by different rules of plasticity in populations [31,32,33,34]. Recent works in this direction have been devoted to the study of networks with asymmetric [35,36] and distance-dependent adaptive connections [37]. The result of the heterogeneity of the rules of adaptive interaction was the emergence of new effects, such as recurrent synchronization and transient circulant clusters. However, the question of the influence of the heterogeneity of adaptive couplings still remains little studied.

The aim of this work is to analyze the effects that arise in oscillatory networks in the case of an heterogeneous character of interactions, which includes several different rules of for the adaptation of couplings and time scales of their change. We study this issue within the framework of a two-population network as the simplest model that provides the implementation of various schemes of heterogeneous interactions at the level of relatively small groups. First, we separately analyze the effects arising from the introduction of heterogeneity only in terms of the type or rate of coupling adaptation. We demonstrate that different schemes of heterogeneous adaptation lead to the formation of transient phase clusters of different types. In particular, the introduction of heterogeneity according to the adaptation rule leads to the formation of transient circulant clusters in populations of a network. Another type of transient synchronous behavior, which we refer to as pulsating clusters, is observed in the case of heterogeneity in the rate of coupling adaptation within and between populations of the network. In the next step, we consider the combined influence of these factors on the previously discovered effects. We show that the inclusion of several types of coupling adaptation heterogeneity can lead to the appearance of transient states with features of both circulant and pulsating clusters.

## 2. Model and Methods

### 2.1. The Model of an Adaptive Oscillatory Network

In this paper, we consider the network of N=100  adaptively coupled identical phase oscillators. The network consists of two populations of identical size between the elements of which various schemes of heterogeneous interactions can be implemented according to the law and rate of coupling adaptation. The dynamics of the oscillators is described by the following Kuramoto–Sakaguchi equation:(1)$$\begin{array}{c} \frac{d\phi_{i}^{p}}{dt}=\omega -\frac{1}{N}\sum_{q=1}^{2}\sum_{j=1}^{N/2}\kappa_{ij}^{pq}\sin\left(\phi_{i}^{p}-\phi_{j}^{q}+\alpha \right), \end{array}$$
where $\phi_{i}^{p}\in \left[0,2\pi \right)$  describes the phase of the *i*th oscillator (i=1,⋯,N/2)  in the population *p*(p=1,2) , and ω  is the natural frequency, which we set as ω=1 . The parameter α  can be considered to be a phase lag of the interaction between oscillators. The coupling weights $\kappa_{ij}^{pq}\in \left[-1,1\right]$  characterize the strength of connection from the *j*th oscillator of the population *q* on the *i*th oscillator in the population *p*. To describe the evolution of the coupling strength $\kappa_{ij}^{pq}$ , we use the model proposed in [38]:(2)$$\begin{array}{c} \frac{d\kappa_{ij}^{pq}}{dt}=-\varepsilon_{pq}\left(\sin\left(\phi_{i}^{p}-\phi_{j}^{q}+\beta_{q}\right)+\kappa_{ij}^{pq}\right). \end{array}$$The rule according to which the coupling strength changes depending on the states of the interacting network nodes, namely the phase values of the corresponding oscillators, is determined by the adaptation function $\Lambda \left(\phi_{i}^{p}-\phi_{j}^{q},\beta_{q}\right)=-\sin\left(\phi_{i}^{p}-\phi_{j}^{q}+\beta_{q}\right)$ , where $\beta_{q}$  is control parameter. Using the parameter $\beta_{q}$ , the adaptation function can take into account various plasticity rules that can be found in neural networks. For example, for β=0 , the adaptation function has the form $\Lambda =-\sin(\phi_{i}^{p}-\phi_{j}^{q})$ . The sign of the function depends on the temporal order of the oscillators. The adaptation function provides strengthening of the coupling $\kappa_{ij}^{pq}$  if the oscillator *j* precedes oscillator *i* and a decrease in the coupling strength in the opposite case. In a number of previous papers [20,23], this type of adaptation was referred as a causal rule. In neuroscience [39,40], such a relationship is typical for spike time-dependent plasticity (STDP). For $\beta_{q}=-\pi /2$ , the adaptation function takes the form $\Lambda =\cos(\phi_{i}^{p}-\phi_{j}^{q})$  that provides relationship of variables qualitatively similar to the Hebbian learning rule used in neuroscience [41]. In this case, the coupling strength $\kappa_{ij}^{pq}$  is increasing between any two oscillators with close phases, i.e., $\phi_{i}^{p}-\phi_{j}^{q}$  close to zero. If $\beta_{q}=\pi /2$ , then the adaptation function $\Lambda =-\cos(\phi_{i}^{p}-\phi_{j}^{q})$ , which has the opposite effect of the Hebbian-like function. In [20], this type of adaptation was referred as the anti-Hebbian rule. Variation of the parameter $\beta_{q}$  in the interval [−π,π]  makes it possible to systematically study the dynamics of the adaptive network (Equation 1) and (Equation 2) without being limited to only the three cases mentioned above. The parameter $\varepsilon_{pq}$  in (Equation 2) characterizes the time scale of the change in the coupling strength between arbitrary oscillators of *p* and *q* populations.

Thus, the nature of interactions within the network is specified by the set of parameters $\beta_{q}$  and $\varepsilon_{pq}$ , (p,q=1,2) . Features of the organization of interactions in the network are schematically presented in Figure 1. According to the above scheme, the network can implement a structure of heterogeneous interaction with different rules of coupling adaptation for elements of one’s own and another’s population. Thus, for the *i*th oscillator of the first population, the change in the coupling strength obeys the adaptation rule $\Lambda_{1}=\Lambda \left(\phi_{i}^{1}-\phi_{j}^{1},\beta_{1}\right)$  with the oscillators of the same population and the rule $\Lambda_{2}=\Lambda \left(\phi_{i}^{1}-\phi_{j}^{2},\beta_{2}\right)$  with the oscillators of the neighboring population. The type of interactions between the oscillators of the corresponding populations is specified by choosing the parameters $\beta_{1}$  and $\beta_{2}$ , which determine the form of the adaptation functions $\Lambda_{1}$  and $\Lambda_{2}$ , respectively. Heterogeneity in the rate of adaptation is set by parameters characterizing the time scales of changes in the coupling strength between oscillators within individual populations $\varepsilon_{pp}$  and between oscillators of different populations $\varepsilon_{pq}$ (p,q=1,2) .

### 2.2. Basic Dynamic States in a Homogeneous Adaptive Network and Methods for Their Identification

The dynamic states of the network (Equation 1) and (Equation 2) in the case of identical adaptive couplings, i.e., $\beta_{1}=\beta_{2}=\beta$  and $\varepsilon_{11}=\varepsilon_{12}=\varepsilon_{21}=\varepsilon_{22}=\varepsilon$ , have been studied in a series of previous works [22,23,25]. To determine the states formed in adaptive networks, we used a technique based on a cooperative analysis of a number of time-averaged basic characteristics proposed in [25]. The calculation of these characteristics was carried out over a large time interval $\Delta t=10^{5}$  skipping a transient time interval *T*. In particular, such characteristics include the time-averaged order parameters given by
(3)$$\langle R_{k}\rangle =\frac{1}{\Delta t}\int_{T}^{T+\Delta t}R_{k}dt,$$
where
$$R_{k}=|\frac{1}{N}\sum_{j=1}^{N}e^{-ik\phi_{j}}|,k=1,2.$$For a homogeneous adaptive network, we omit the superscript *p* for the phases $\phi_{j}$  in (Equation 3) and sum over all the oscillators of the network (j=1,…,N) . The characteristics $\langle R_{1}\rangle$  and $\langle R_{2}\rangle$  can take values in the interval [0,1]  and indicate the formation of a complete synchronization mode or two synchronized groups with an antiphase relationship if they are an equal unit, respectively. If $0<\langle R_{k}\rangle <1$ (k=1,2) , the network shows more complex behavior whose analysis requires additional characteristics. To determine these characteristics we calculate the degree of mutual synchronization for each pair of oscillators *i* and *j*
(4)$$R_{ij}=|\frac{1}{\Delta t}\int_{T}^{T+\Delta t}e^{i(\phi_{i}\left(t\right)-\phi_{j}\left(t\right))}dt|.$$The value of each element in the resulting matrix R  is bounded in the interval [0,1] , being $R_{ij}=1$  when the oscillators *i* and *j* are synchronized, i.e., phase difference $\phi_{i}\left(t\right)-\phi_{j}\left(t\right)=const$ . Once the degree of synchronization between the oscillators is measured, we construct a new matrix $\tilde{R}$  whose elements take a value ${\tilde{R}}_{ij}=1$  when oscillators *i* and *j* are synchronized or ${\tilde{R}}_{ij}=0$  otherwise. Based on the obtained matrix $\tilde{R}$ , we introduce two other characteristics to make it possible to distinguish complex synchronous modes, such as multicluster and chimera states. One of these describes the fraction of synchronized pairs of oscillators between which there is a connection which is defined as follows:(5)$$\begin{array}{c} R_{link}=\frac{1}{2N_{L}}\sum_{i,j=1}^{N}{\tilde{R}}_{ij}, \end{array}$$
where $N_{L}$  is the total number of connections between oscillators in the network. In particular, this characteristic makes it possible to detect the coherent state of the network with a fixed phase relationship between the oscillators. In this state, all oscillators of the network are frequency-synchronized $\left(R_{link}=1\right)$ , and the phases of the oscillators are almost randomly distributed without forming clusters $\left(\langle R_{1}\rangle \approx 0,\langle R_{2}\rangle \approx 0\right)$ . If the parameter $R_{link}\in \left(0,1\right)$ , this may indicate the formation of either several synchronous groups with different frequencies (*M* frequency clusters) or a chimera state. To distinguish between these states, we introduce the characteristic $P_{C}$  defined as follows
(6)$$\begin{array}{c} P_{C}=\frac{1}{N}\sum_{i=1}^{N}\max_{j}\left\{{\tilde{R}}_{ij}\right\}. \end{array}$$This characterizes the fraction of network oscillators forming synchronous clusters. To find this parameter, we actually calculate the number of columns (or rows) of the matrix $\tilde{R}$  whose elements contain nonzero values. The network is divided to *M* frequency clusters if the parameters (Equation 5) and (Equation 6) satisfy the conditions $R_{link}\in \left(0,1\right),P_{C}=1$ . Moreover, the number of formed clusters *M* is inversely proportional to the value of $R_{link}$ . The fulfillment of the conditions $P_{C}\in \left(0,1\right)$  indicates the formation of a chimera state in the network. In this case, the quantity $P_{C}$  characterizes the size of the coherent part of the chimera state. The dynamics of the oscillator phases in the case of such complex multicluster modes can be estimated based on the values of the characteristics $\langle R_{k}\rangle$ (k=1,2) . As an example, consider the dependencies of the basic characteristics on the parameter β  presented in Figure 2a. The characteristics are obtained for the values of the parameters α=0.3π , ε=0.01  and a fixed set of initial conditions chosen randomly with a uniform phase distribution $\phi_{i}$  in the interval [0,2π]  and coupling strengths $\kappa_{ij}$  in [−1,1] . These dependencies illustrate the influence of the rule of coupling adaptation on the properties of the states formed in the network. It is shown that a homogeneous adaptive network exhibits a wide range of different synchronous states, including chimera ones [22]. Synchronous states include one-cluster (M=1)  and multicluster (M>1)  modes of frequency synchronization. In the case of one-cluster states, the temporal behavior of frequency synchronized oscillators can be defined as
(7)$$\begin{array}{c} \phi_{i}\left(t\right)=\Omega t+\chi_{i},\;\;\;i=1,\cdots ,N, \end{array}$$
where Ω  is a collective frequency, and $\chi_{i}\in \left[0,2\pi \right)$  describes individual phase shifts. Depending on the type of coupling adaptation, the distribution of oscillator phases within the frequency cluster corresponds to one of the two main types of behavior: antipodal and splay (Figure 2).

In the case of the antipodal type of behavior, the oscillators are divided into two groups (clusters $C_{1}$  and $C_{2}$  in Figure 2b), which are in antiphase with respect to each other, i.e., $\chi_{i}\in \left\{\chi ,\chi +\pi \right\}$  with χ∈[0,2π) . Such state is characterized by $\langle R_{2}\rangle =1$  for one frequency cluster (M=1)  in the network. Note that the global synchronization mode is a particular case of such a mode when the size of one of the antiphase clusters is equal to zero. Accordingly, the domains of existence of these modes in the parameter space also coincide. However, the global synchronization mode is observed when the initial conditions are appropriately specified. For example, it can be obtained by choosing $\kappa_{ij}\left(0\right)$  in the range of positive values. Therefore, the global synchronization mode is not explicitly indicated in Figure 2. States containing *M*(M>1)  frequency clusters of antipodal type are characterized by relatively large values of $\langle R_{2}\rangle \in \left(0,1\right)$ . At the same time, other basic characteristics take the values $R_{link}\in \left(0,1\right),P_{C}=1$  (see Figure 2a). In this case, the main contribution to the value of $\langle R_{2}\rangle$  is determined by the synchronous group of the largest size [22]. Thus, a decrease in the values $\langle R_{2}\rangle$  and $R_{link}$  in Figure 2a indicates an increase in the number *M* of frequency clusters of the antipodal type. The synchronous behavior of the antipodal type is supported by Hebbian-like adaptation (the vicinity of the point β=−π/2 ).

In the synchronous clusters of the splay type (Figure 2c), the phases are almost uniformly distributed across the interval $\chi_{i}\in \left[0,2\pi \right)$ , such that the order parameters $\langle R_{k}\rangle ,\left(k=1,2\right)$  take values close to zero. As shown in Figure 2a, splay states are characterized by $R_{link}=1,P_{C}=1$  for one frequency cluster (M=1)  in the network and $R_{link}\in \left(0,1\right),P_{C}=1$  in case of multiclusters (M>1) . The splay states are mainly realized in the case of STDP-like rule of coupling adaptation (the vicinity of the point β=0  in Figure 2a).

The classification of the dynamical states of a homogeneous adaptive network based on the characteristics (Equation 3)–(Equation 6) is presented in the Table 1. When analyzing states in a two-population network, we calculate these characteristics for individual populations and the network as a whole. The information summarized in Table 1 makes it possible to identify the states typical for a homogeneous network and to identify new ones in the presence of anomalous values of characteristics.

Note that along with the described modes, the network can also demonstrate a family of one-frequency cluster states. Such states are characterized by the presence of several phase clusters, the phase relationships between which depend on the number and size of clusters formed. An example of such a state is shown in Figure 2d. In this case, there are four phase clusters. Two of them (clusters $C_{1}$  and $C_{2}$ ) are in antiphase, and the rest ($C_{3}$  and $C_{4}$ ) have phase shifts ψ  and φ  relative to the $C_{1}$  cluster. The technique for finding the phase relationships between clusters for one-frequency states will be described in detail in Section 3.1. We will refer to such states as being generalized. Numerical experiments have shown that one-frequency states of the generalized type have small basins of attraction and are realized when initial conditions are specified in a small neighborhood of the considered solutions. A relatively weak perturbation of the initial conditions leads to a change in the behavior of the network associated with the transition to another attractor. One-frequency states of the generalized type are observed in the domain of existence of the splay states. Due to the smallness of the basins of attraction, these modes could not be detected in the course of previous numerical experiments. It can be expected that the introduction of heterogeneity of intranetwork interactions can affect the conditions for the existence of these cluster states and, as a consequence, lead to the emergence of new effects. Therefore, in studying the influence of the heterogeneous nature of adaptive couplings, first of all, we will analyze cases where the rules of adaptation of one or both populations correspond to the region of existence of the splay states.

## 3. Results

### 3.1. Effects in a Network with Heterogeneity according to the Rule of Coupling Adaptation

In this section, we analyze the dependence of the dynamic states of the network (Equation 1) and (Equation 2) on the implemented rules of coupling adaptation when their time scales are identical $\varepsilon_{11}=\varepsilon_{22}=\varepsilon_{12}=\varepsilon_{21}=\varepsilon$ . We also assume that the dynamics of couplings is slower than the phase dynamics, and this is reflected by a small parameter 0<ε≪1 . The results of the study are presented as a two-parameter diagram of the dynamical states in Figure 3. Note that in the diagram, we have shown the regions of existence of only individual dynamic states that play a role in the formation of new effects. Along the edges of the diagram, there are dependencies of the state characteristics of the network in a homogeneous case (as in Figure 2a), illustrating the relationship between the adaptation rule and the type of network behavior. This makes it possible to trace the influence of combining adaptation rules that provide different types of synchronous behavior in a homogeneous case at the level of a heterogeneous two-population network.

#### 3.1.1. Combination of Adaptation Rules Supporting Splay States

Consider the behavior of the network (Equation 1) and (Equation 2) with a combination of adaptation rules corresponding to splay states in a homogeneous network, i.e., when the parameters $\beta_{p}\in \left[-0.4\pi ,0.4\pi \right]$ , p=1,2 . In this case, the introduction of heterogeneity according to the adaptation rule $\left(\beta_{1}\neq \beta_{2}\right)$  leads to the loss of stability of synchronous states of the splay type. Instead of splay states, a set of one-frequency cluster states appears in this parameter range. As an example, Figure 3 shows the phase distributions of oscillators and the domains of existence for only three typical cluster states. We classify these modes as generalized cluster states by analogy with Section 2.2. Such states differ in the size and number of phase clusters in the populations of the network, as well as in the relative phase shifts between them.

Let us find the conditions for the distribution of oscillator phases for generalized cluster states using the example of the state shown in Figure 3a. In this case, the first population contains two antiphase clusters $C_{1}$  and $C_{2}$ , and the second population includes three phase clusters $C_{3}$ , $C_{4}$  and $C_{5}$  having sizes $N_{C_{i}}$ , i=1,⋯,5 . The temporal behavior for the frequency-synchronized oscillator can be presented as follows:(8)$$\begin{array}{c} \phi_{i}^{p}\left(t\right)=\Omega t+\chi_{i}^{p},\;\;\;i=1,\ldots ,N/2,\;p=1,2, \end{array}$$
where Ω  is the collective frequency of oscillators in the network. For the corresponding clusters in populations, the phases $\chi_{i}^{p}$  take the following values $\chi_{i}^{1}\in \left\{\chi ,\chi +\pi \right\}$ , $\chi_{i}^{2}\in \left\{\chi +\psi ,\chi +\psi +\pi ,\chi +\varphi \right\}$  with χ,ψ,φ∈[0,2π) . Substituting (Equation 8) in (Equation 2), we obtain stationary values for couplings
(9)$$\begin{array}{c} \kappa_{ij}^{pq}=-\sin\left(\chi_{i}^{p}-\chi_{j}^{q}+\beta_{q}\right), \end{array}$$
where (i,j=1,⋯,N/2) , (p,q=1,2) . Taking into account (Equation 8) and (Equation 9), from Equation (Equation 1), we find expressions for the collective frequency of oscillators belonging to the $C_{1}$ , $C_{3}$  and $C_{5}$  clusters, respectively
(10)$$\begin{array}{c} \Omega =\omega +\frac{1}{N}\left[\left(\frac{N}{2}-1\right)\sin\left(\alpha \right)\sin\left(\beta_{1}\right)+\left(\frac{N}{2}-N_{C_{5}}\right)\sin\left(\psi -\alpha \right)\sin\left(\psi -\beta_{2}\right)+N_{C_{5}}\sin\left(\varphi -\alpha \right)\sin\left(\varphi -\beta_{2}\right)\right],\\ \Omega =\omega +\frac{1}{N}\left[\frac{N}{2}\sin\left(\psi +\alpha \right)\sin\left(\psi +\beta_{1}\right)+\left(\frac{N}{2}-N_{C_{5}}-1\right)\sin\left(\alpha \right)\sin\left(\beta_{2}\right)+N_{C_{5}}\sin\left(\psi -\varphi +\alpha \right)\sin\left(\psi -\varphi +\beta_{2}\right)\right],\\ \Omega =\omega +\frac{1}{N}\left[\frac{N}{2}\sin\left(\varphi +\alpha \right)\sin\left(\varphi +\beta_{1}\right)+\left(\frac{N}{2}-N_{C_{5}}\right)\sin\left(\psi -\varphi +\alpha \right)\sin\left(\psi -\varphi +\beta_{2}\right)+\left(N_{C_{5}}-1\right)\sin\left(\alpha \right)\sin\left(\beta_{2}\right)\right]. \end{array}$$Note that the expressions for the collective frequency of oscillators belonging to antiphase clusters will have the same form. By equating the right-hand sides of Equation (Equation 10), we obtain a system of two equations for the phase shifts ψ  and φ  which can be written in the following form:(11)$$\begin{array}{c} \tan\left(\alpha \right)\tan\left(b_{-}\right)\left(1+\tan^{2}\left(\psi \right)\right)\left(1+\tan^{2}\left(\varphi \right)\right)+\frac{N}{2}\left(1+\tan^{2}\left(\varphi \right)\right)\tan\left(\psi \right)\left(1-\tan\left(b_{-}\right)\tan\left(\psi \right)\right)\left(\tan\left(\alpha \right)+\tan\left(b_{+}\right)\right)+\\ +N_{C_{5}}\tan\left(\psi \right)\left(\tan\left(\psi \right)-\tan\left(\varphi \right)\right)[1-\tan\left(\alpha \right)\tan\left(b_{+}\right)+\tan\left(b_{-}\right)\left(\tan\left(\alpha \right)+\tan\left(b_{+}\right)\right)+\tan\left(\varphi \right)(\tan\left(\alpha \right)+\tan\left(b_{+}\right)-\\ -\tan\left(b_{-}\right)+\tan\left(\alpha \right)\tan\left(b_{+}\right)\tan\left(b_{-}\right))]=0,\\ \tan\left(\varphi \right)[\left(1-\tan\left(\alpha \right)\tan\left(b_{+}\right)\right)\left(\frac{N}{2}-N_{C_{5}}-\frac{N}{2}\tan\left(b_{-}\right)\tan\left(\psi \right)\right)-N_{C_{5}}\tan\left(b_{-}\right)\left(\tan\left(\alpha \right)+\tan\left(b_{+}\right)\right)]-\\ -\tan\left(\psi \right)\left[\left(\frac{N}{2}-N_{C_{5}}\right)\tan\left(b_{-}\right)\left(\tan\left(\alpha \right)+\tan\left(b_{+}\right)\right)-N_{C_{5}}\left(1-\tan\left(\alpha \right)\tan\left(b_{+}\right)\right)\right]-\frac{N}{2}\left(\tan\left(\alpha \right)+\tan\left(b_{+}\right)\right)=0, \end{array}$$
where $b_{\pm }=\left(\beta_{1}\pm \beta_{2}\right)/2$ . Thus, we have obtained a family of generalized cluster states with different cluster sizes $N_{C_{i}}$ , i=1,⋯,5  and relative phase shifts ψ  and φ  satisfying the system of Equation (Equation 11). Similarly, one can obtain conditions for the generalized cluster states shown in the Figure 3b,c.

The appearance of such modes is a consequence of an increase in the basins of attraction of one-frequency generalized cluster states (see Section 2.2) caused by the introduction of heterogeneity according to the rules of coupling adaptation. Thus, the considered range of values of the parameters $\beta_{1}$  and $\beta_{2}$  is characterized by high multistability due to the coexistence of a family of one-frequency generalized cluster states. A consequence of the emergence of multistable network properties near the homogeneous case ($\beta_{1}$  = $\beta_{2}$ ) is the appearance of a new effect associated with the formation of transient circulant clusters [35]. The properties of the network behavior in this state are illustrated in Figure 4. In such states, in both populations of the network, we observed antiphase clusters (Figure 4c), the size and composition of which slowly changes over time. The process of rearrangement clusters consists in successive transitions of oscillators from one cluster to another (Figure 4b). Such transitions occur in a certain direction and in a given order of the oscillators (Figure 4a). The region of existence of circulant clusters is highlighted by double hatching in the diagram (Figure 3d).

#### 3.1.2. Combination of Adaptation Rules Supporting States of Antipodal Type

Now consider the behavior of the network (Equation 1) and (Equation 2) with a combination of adaptation rules corresponding to states of antipodal type in a homogeneous network ($\beta_{q}\in \left[-\pi ,-0.4\pi \right]$ , q=1,2 ). In this case, the introduction of heterogeneity according to the type of adaptation leads to the formation of one-frequency generalized cluster states with certain properties. An example of a typical distribution of oscillator phases in such a state is shown in Figure 3c. The network contains four phase clusters: two pairs of antiphase clusters shifted relative to each other by a fixed phase ψ . To find the condition for the value of ψ , one can use the representation of the solution for a one-frequency generalized cluster state in the form (Equation 8), (Equation 9), where the phases take the values $\chi_{i}^{1}\in \left\{\chi ,\chi +\pi \right\}$ , $\chi_{i}^{2}\in \left\{\chi +\psi ,\chi +\psi +\pi \right\}$  with χ,ψ∈[0,2π) . Similarly to Section 3.1.1, using the solution (Equation 8) and (Equation 9) from the equation (Equation 1), we find expressions for the collective frequency of oscillators in the first and second population, respectively:(12)$$\begin{array}{c} \Omega =\omega +\frac{1}{N}\left[\left(\frac{N}{2}-1\right)\sin\left(\alpha \right)\sin\left(\beta_{1}\right)+\frac{N}{2}\sin\left(\psi -\alpha \right)\sin\left(\psi -\beta_{2}\right)\right],\\ \Omega =\omega +\frac{1}{N}\left[\frac{N}{2}\sin\left(\psi +\alpha \right)\sin\left(\psi +\beta_{1}\right)+\left(\frac{N}{2}-1\right)\sin\left(\alpha \right)\sin\left(\beta_{2}\right)\right]. \end{array}$$Equating the right-hand sides of the expressions for the collective frequency Ω  in Equation (Equation 12), we can ascertain that the value of the phase shift ψ  between clusters of different populations satisfies the following equation:(13)$$\begin{array}{c} \sin\left(2\psi +b_{-}\right)\sin\left(\alpha +b_{+}\right)=\sin\left(b_{-}\right)\left(\sin\left(b_{+}-\alpha \right)+\left(2-\frac{4}{N}\right)\sin\alpha \cos\left(b_{+}\right)\right), \end{array}$$
where $b_{\pm }=\left(\beta_{1}\pm \beta_{2}\right)/2$ . In [26], this type of synchronous behavior was referred to as double antipodal. In what follows, we will also use this notation for such states. It was indicated as the third type of one-cluster (frequency cluster) states of a homogeneous adaptive network but was not detected in numerical experiments [23,26] due to the smallness of the basin of attraction. In a heterogeneous network, such states exist in a wide range of values of the $\beta_{1}$  and $\beta_{2}$  parameters, not being limited only by the adaptation rules corresponding to the behavior of the antipodal type (Figure 3d).

#### 3.1.3. Combination of Adaptation Rules Supporting States of Antipodal and Splay Types

In this section, we briefly describe the effects observed in the network (Equation 1) and (Equation 2) with a combination of adaptation rules corresponding to states of different types (antipodal and splay) in a homogeneous network, i.e., when $\beta_{1}\in \left[-0.4\pi ,0.4\pi \right]$ , $\beta_{2}\in \left[-\pi ,-0.4\pi \right]$  or $\beta_{1}\in \left[-\pi ,-0.4\pi \right]$ , $\beta_{2}\in \left[-0.4\pi ,0.4\pi \right]$ . In this case, the network can also exhibit many different generalized cluster states, as shown in the diagram (Figure 3d). Such a combination of adaptation rules can also lead to the appearance of transient circulant clusters in only one of the populations of the network. At the same time, in another population of the network, we observe stationary antiphase clusters whose properties do not change in time. In the diagram (Figure 3d), the regions of existence of such modes are highlighted by single shading. As shown in the Figure 3d, this behavior is realized at the boundary of the stability region of generalized cluster states. The properties of transient circulant clusters are studied in detail in [35].

### 3.2. Effects Observed in a Network with Heterogeneity in the Rate of Coupling Adaptation

In this section, we consider the influence on the collective behavior of the network (Equation 1) and (Equation 2) of heterogeneity only in the rate of coupling adaptation. We assume that $\beta_{1}=\beta_{2}=\beta$ . An analysis of various schemes for organizing heterogeneity in terms of the rate of coupling adaptation within individual populations and between them showed that the properties of the collective behavior of the network are dominated by the choice of the rule of adaptation. Therefore, we will discuss only the case when the introduction of heterogeneity in the rate of coupling leads to the appearance of new effects. In particular, this is observed in the case when the coupling strength between populations changes faster than when between oscillators within individual populations. Next, we will consider how the network dynamics changes depending on the parameter β  for the following distribution of the rates of coupling adaptation: $\varepsilon_{11}=\varepsilon_{22}=0.01$ , $\varepsilon_{12}=\varepsilon_{21}=0.1$ .

#### 3.2.1. Changing the Type of Synchronous States

The behavior of the network with this scheme of heterogeneous interactions is characterized by one-parameter diagrams of basic characteristics (Equation 3)–(Equation 6) for the individual populations, as shown in Figure 5a. A comparative analysis of the diagrams shown in Figure 2a and Figure 5a indicates that significant changes in the network behavior are observed in the range of values of the parameter β∈[−0.4π,0.5π] . In the case of a homogeneous network, this area corresponds to states of the play type. In particular, when passing through the value β=0 , the type of the synchronous state changes smoothly from splay to antipodal, which is not observed in a homogeneous adaptive network. As the parameter β  increases, antiphase clusters begin to gradually form in each of the populations. This confirms the smooth growth of the time-averaged order parameters $\langle R_{2}^{p}\rangle$ , (p=1,2)  from 0 to 1 (Figure 5a). The superscript *p* indicates the number of the population characterized by the parameter. As a result, a synchronous state of the double antipodal type with ψ=π/2  is formed in the network.

Next, we give an explanation of the resulting effect. Since couplings within populations change more slowly than between them, it is the interpopulation interaction that has a predominant effect on the collective dynamics of the network. This assumption was confirmed numerically. In a number of simulations, we found that the behavior of a network in which slow adaptive couplings within populations are replaced by constant coupling weights with random values $k_{ij}^{pp}\in \left[-1,1\right]$ , (p=1,2) , (i,j=1,⋯,N/2)  coincides with the dynamics of the original network. Therefore, the transition from one type of behavior to another can be explained on the basis of a simple two-oscillator model, replacing each population with an oscillator. A detailed analysis of the dynamics of two phase oscillators with adaptive couplings was carried out in [38]. In this range of parameters, two adaptively coupled oscillators have two stable synchronous modes, characterized by a phase difference $\phi_{1}-\phi_{2}=\pm \pi /2$ . We also observe similar values of phase shifts between clusters of different populations of the original network (Equation 1) and (Equation 2). The situation when arbitrary pairs of oscillators from different populations satisfy the condition $\phi_{i}^{1}-\phi_{j}^{2}=\pm \pi /2$  is realized only if there are relationships $\phi_{i}^{p}-\phi_{j}^{p}=\chi$ , where χ∈{0,π} , i,j=1,⋯,N/2  within populations. This phase relation corresponds to the synchronous mode of the double antipodal type with ψ=π/2 . The loss of stability of this synchronous mode in the two-oscillator model also leads to the destruction of the double antipodal state in the network (Equation 1) and (Equation 2).

#### 3.2.2. Transient Pulsating Clusters

The destruction of this mode is characterized by anomalous values of the basic characteristics (Equation 3)–(Equation 6) (shaded area in Figure 5a). In this case, the order parameters $\langle R_{2}^{p}\rangle$  take on sufficiently large values, while the remaining characteristics $\left(R_{link}^{p}=0,P_{C}^{p}=0\right)$  indicate a lack of synchronization in the network. It has been established that such behavior of the characteristics is explained by the emergence of a new type of transient synchronous behavior in the network. The behavior of the network is an alternation of intervals of asynchronous behavior of the network and the existence of synchronous states of the double antipodal type. The features of this process are illustrated in Figure 5b–e. The evolution of the order parameters $\langle R_{2}^{p}\rangle$  of individual populations (Figure 5d) demonstrates time intervals of constant values of $\langle R_{2}^{p}\rangle =1$  and areas of a rather sharp change in the parameters up to zero values. This indicates the process of alternation of synchronous and asynchronous stages of the network behavior, respectively. At the same time, the processes of the emergence and destruction of synchronous states in both populations are triggered simultaneously. The behavior of oscillators in this state is demonstrated by the time dependencies of the relative phase differences of two arbitrarily chosen oscillators in each of the populations (Figure 5b–d). These dependencies, as well as the distribution of phase values (Figure 5c) of two arbitrary oscillators of the network at $t=10^{5}$  show that during the synchronization intervals, a pair of antiphase clusters is formed in each population of the network. At each new stage of network synchronization, the set of oscillators that form the corresponding clusters change randomly. The process of their rearrangement is illustrated by the space–time diagram in Figure 5e. To visualize these changes, we reordered the indices of the oscillators in accordance with their belonging to the corresponding phase clusters at the initial moment of time. In subsequent synchronization intervals, the distribution of oscillators among clusters was carried out relative to the first oscillator of population p=1 , assuming that it always belongs to cluster $C_{1}$ . We refer to this behavior of the network as transient pulsating clusters.

### 3.3. Effects Observed in the Network with the Combined Organization of Heterogeneity according to the Rules and the Rate of Coupling Adaptation

In this section, we consider the combined influence of the heterogeneity of interactions according to the rule and the rate of coupling adaptation. In particular, we analyze how such an organization of interactions affects the new effects found in the presence of only one of the factors in Section 3.1 and Section 3.2. We studied the behavior of the network depending on the parameters $\beta_{1}$  and $\beta_{2}$  for fixed values $\varepsilon_{11}=\varepsilon_{22}=0.01$ , $\varepsilon_{12}=\varepsilon_{21}=0.1$ . The results of this analysis are presented in the form of a state diagram in Figure 6. The diagram contains only the areas of existence of modes associated with the emergence of new effects of network behavior. A comparative analysis of the diagrams in Figure 6d and Figure 3d shows that the introduction of several heterogeneity factors retains the possibility of the existence of a set of generalized cluster states and also transient states described in Section 3.1 and Section 3.2. The conditions of their existence are also preserved. The mode of circulant clusters in both populations of the network exists in the vicinity of the case of identical adaptation rules (the area with double hatching in Figure 6d). A combination of rules corresponding to different types of behavior is required for the formation of circulant clusters in only one of the populations (the area with horizontal hatching in Figure 6d).

Another area with transient behavior is highlighted by oblique hatching in the diagram (Figure 6d). If the adaptation rules are similar ($\beta_{1}\approx \beta_{2}$ ), we observe pulsating clusters, which are described in Section 3.2. An increase in the mismatch of the parameters $\beta_{1}$  and $\beta_{2}$  inside this region leads to the appearance of transient synchronous states, the properties of which are typical for both of the heterogeneity mechanisms considered earlier. The network behavior properties in this case are illustrated in Figure 7. The evolution of the network state includes several successive stages (Figure 7a). The first stage (slow rebuilding) corresponds to the existence of the synchronous state of the double antipodal type (Figure 7d). During the second stage (fast rebuilding), we observe the rebuilding of clusters (Figure 7e) accompanied by the appearance of a second frequency group (designated as $\Omega_{II}$  in Figure 7c) of oscillators in the second population of the network. The space–time diagram in Figure 7b shows, in detail, the process of rebuilding clusters in the network. The transition from the slow stage to the fast one is initiated by the transition of one of the oscillators of the first population from one cluster to another. This starts the process of circulant rearrangements of oscillators in a given population, the speed of which increases (Figure 7b). Changes in the first population lead to a rebuilding of the clusters in the second population, which is associated with the successive transition of a certain number of oscillators into asynchronous behavior. The exit of these oscillators from the synchronization behavior is associated with the suppression of couplings with the rest of the network oscillators. Thus, an asynchronous group (group $\Omega_{II}$ ) is formed with frequencies close to the value of the natural frequencies of the oscillators (Figure 7c). The asynchronous group increases as the rate of circulant transitions increases in the first population (Figure 7b). The stage of fast rebuilding ends with an avalanche-like desynchronization of the entire network at t=  375,000. At this moment, the order parameters $R_{2}^{p}$  of both populations decrease almost to zero (Figure 7a). A short-term interval of asynchronous behavior is replaced by a fairly fast synchronization of network oscillators. As a result, the synchronous state of the double antipodal type are newly formed in the network, and the next stage of slow network restructuring follows.

With a relatively large mismatch of the rules of coupling adaptation (on the border of the area with oblique hatching in Figure 6d), the stages of slow and fast rebuilding become irregular. This is determined by a wide variation in the duration of these stages. Another feature is that some of the oscillators can remain in an asynchronous state after the completion of the fast network rebuilding stage.

## 4. Discussion

We have studied the features of synchronous behavior in a two-population network with a heterogeneous character of interactions, either by the rule of coupling adaptation or by the rate of their adaptation. It is apparent that in both cases, the introduction of heterogeneity of interactions leads to the emergence of new modes of collective behavior. In particular, the introduction of heterogeneity according to the communication adaptation rule leads to the loss of stability of one of the two main types of synchronous behavior (splay type) in a homogeneous adaptive network. Instead of splay states, a family of one-frequency cluster modes appears, which coexist in a wide range of parameter values that control the adaptation rules. These cluster states differ in the number, size and relative phase shifts of clusters formed in populations. The appearance of a region of high multistability, along with the destruction of splay states in the vicinity of the homogeneous case, is the reason for the emergence of transient synchronous behavior in the form of the circulant clusters described in [35]. In such states, each population contains a pair of antiphase clusters whose size and composition slowly change over time as result of successive transitions of oscillators between clusters.

Another type of transitive synchronous behavior, which we refer to as pulsating clusters, was discovered when implementing the corresponding scheme of heterogeneity in the rate of coupling adaptation. This behavior could be observed in the case when the couplings between the oscillators of different populations changed faster than when within individual populations. The mode of pulsating clusters represents a sequential alternation of stages of synchronous and asynchronous network behavior.

The implementation of heterogeneity both in type and in the rate of change of adaptive couplings retains the possibility of the existence of transient clusters observed in the presence of only one of the heterogeneity factors and also leads to the appearance of transient synchronous states that have features of both circulant and pulsating clusters. This type of transient behavior is a cyclic sequential alternation of several stages of the evolution of the synchronous behavior of the network and a short interval of complete desynchronization of the oscillators. The stages of synchronous behavior include circulant rearrangements of clusters in populations, i.e., transitions of oscillators between clusters occurring in a certain direction and sequence.

In conclusion, we note that the presented results expand the understanding of the diversity of dynamic behavior and the mechanisms of its occurrence in adaptive networks. The results obtained in the framework of a fairly simple network model show that the heterogeneous adaptation can be one of the mechanisms underlying the formation of coherent metastable states. Such states are important, for example, in neural networks. Neurophysiological experiments [42,43,44] show that the dynamics of brain neural networks is dominated by processes characterized by the appearance of short-term patterns of synchronous activity and spontaneous transitions between them, observed even in the absence of external influence. Understanding the principles of the formation and evolution of such coherent metastable states is important since they underlie the functioning of brain neural networks associated with the performance of cognitive functions.

## Figures and Tables

**Figure 1 entropy-25-00913-f001:**
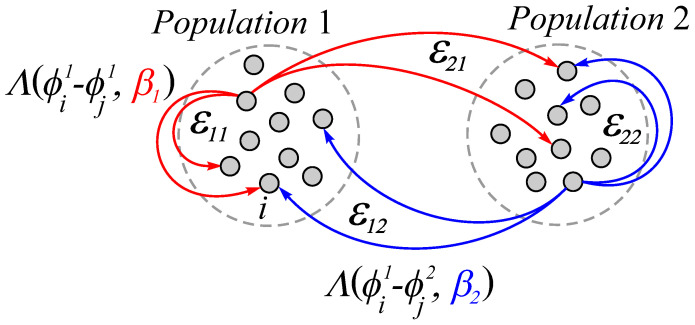
Schematic structure of the heterogeneous interaction in type and rate of adaptation for the two-population network (Equation 1) and (Equation 2). The corresponding adaptation rules implemented within and between populations are highlighted in color.

**Figure 2 entropy-25-00913-f002:**
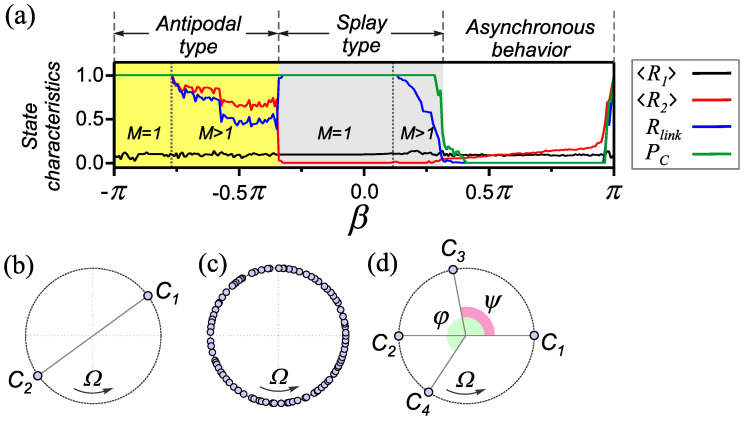
One-parameter diagram and possible types of one-cluster states for a homogeneous adaptive network. (**a**) The dependencies of the characteristics (Equation 3)–(Equation 6) on the parameter β  in a homogeneous adaptive network (Equation 1) and (Equation 2) for α=0.32π , $\beta_{1}=\beta_{2}=\beta$  and $\varepsilon_{11}=\varepsilon_{12}=\varepsilon_{21}=\varepsilon_{22}=0.01$ . The regions of different types of synchronous behavior are identified in accordance with Table 1. Types of one-frequency states: (**b**) antipodal state, (**c**) splay state and (**d**) generalized state.

**Figure 3 entropy-25-00913-f003:**
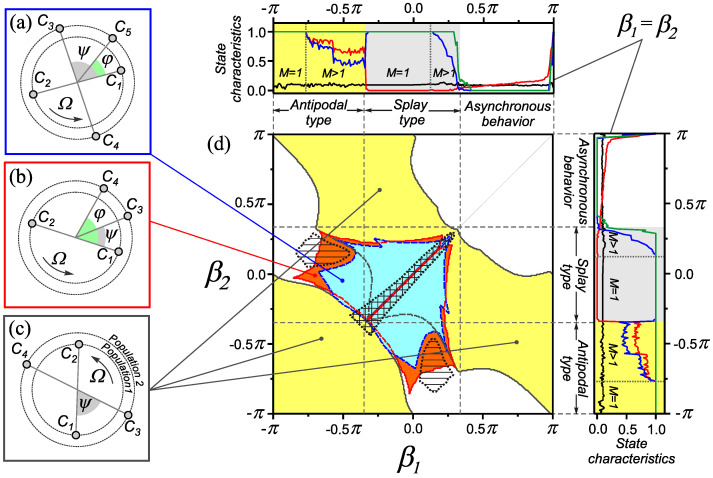
Examples of one-frequency generalized cluster states for the adaptive network (Equation 1) and (Equation 2) and regions of their existence on the parameter plane $\left(\beta_{1},\beta_{2}\right)$ . Oscillator phase distributions for one-frequency generalized cluster states with clusters of various sizes: (**a**) $N_{C_{1}}=35$ , $N_{C_{2}}=15$ , $N_{C_{3}}=25$ , $N_{C_{4}}=24$ , $N_{C_{5}}=1$ ; (**b**) $N_{C_{1}}=30$ , $N_{C_{2}}=20$ , $N_{C_{3}}=35$ , $N_{C_{4}}=15$ ; (**c**) $N_{C_{1}}=25$ , $N_{C_{2}}=25$ , $N_{C_{3}}=40$  and $N_{C_{4}}=10$ . (**d**) The diagram of dynamical states in the parameter plane $\left(\beta_{1},\beta_{2}\right)$ . Along the edges of the diagram, the dependencies of the characteristics (Equation 3)–(Equation 6) for the homogeneous case $\left(\beta_{1}=\beta_{2}\right)$  are shown. These dependencies are similar to those shown in Figure 2a. Parameter values: α=0.32π , $\varepsilon_{11}=\varepsilon_{12}=\varepsilon_{21}=\varepsilon_{22}=0.01$ .

**Figure 4 entropy-25-00913-f004:**
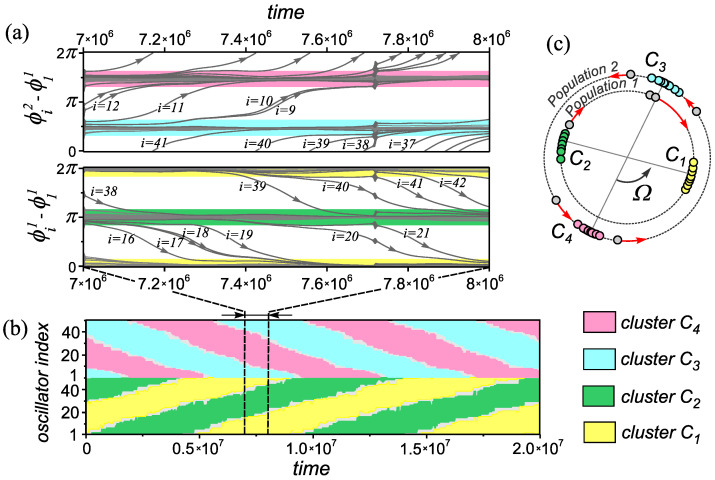
The properties of transient circulant clusters the network (Equation 1) and (Equation 2). (**a**) Evolution of the relative phase differences of oscillators in different populations. (**b**) Spatiotemporal diagrams illustrating the restructuring of clusters in populations. (**c**) Snapshot of the distribution of the phases of the network oscillators at a moment of time $t=7.3\times 10^{6}$ . Parameter values: α=0.32π , $\beta_{1}=-0.23\pi$ , $\beta_{2}=-0.3\pi$  and $\varepsilon_{11}=\varepsilon_{22}=\varepsilon_{21}=\varepsilon_{22}=0.01$ .

**Figure 5 entropy-25-00913-f005:**
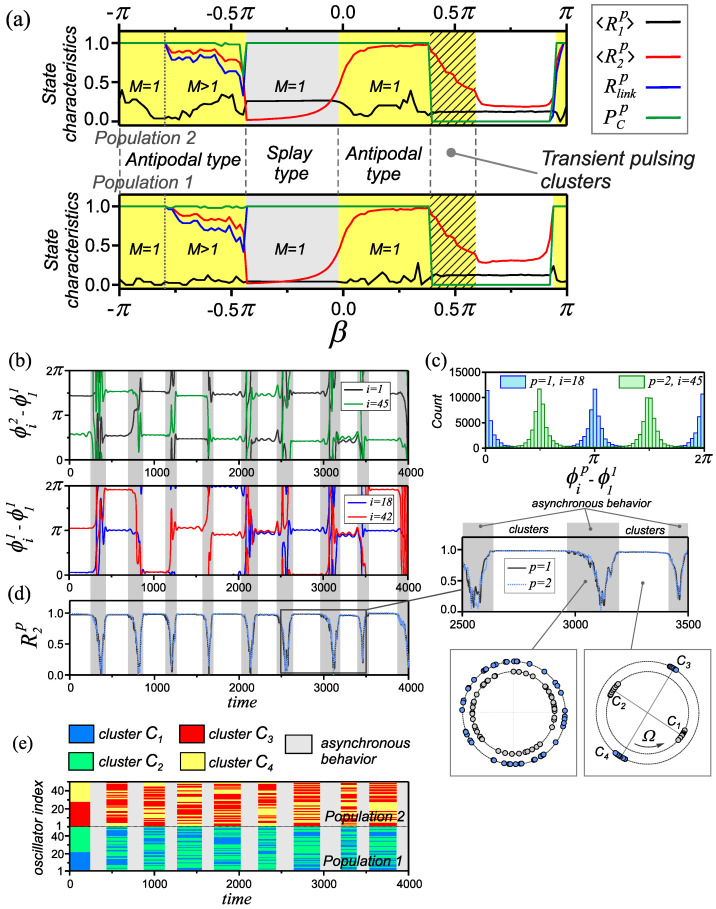
The properties of transient pulsing clusters. (**a**) One–parameter diagrams of characteristics (Equation 3)–(Equation 6) for individual populations of the network (Equation 1), (Equation 2) depending on the parameter $\beta =\beta_{1}=\beta_{2}$ . (**b**) Time dependencies of relative phase differences for a pair of arbitrary oscillators from each population. (**c**) A typical distribution of the values of the phase differences of the oscillators $\phi_{i}^{p}-\phi_{1}^{1}$  over the time interval $t=10^{5}$ . (**d**) The evolution of the order parameters $R_{2}^{p}$ , p=1,2 . (**e**) Spatiotemporal diagrams illustrating the restructuring of the composition of pulsating clusters. Fragments (**b**–**e**) are built for the parameter β=0.41π . Parameter values: α=0.32π , $\varepsilon_{11}=\varepsilon_{22}=0.01$  and $\varepsilon_{21}=\varepsilon_{22}=0.1$ .

**Figure 6 entropy-25-00913-f006:**
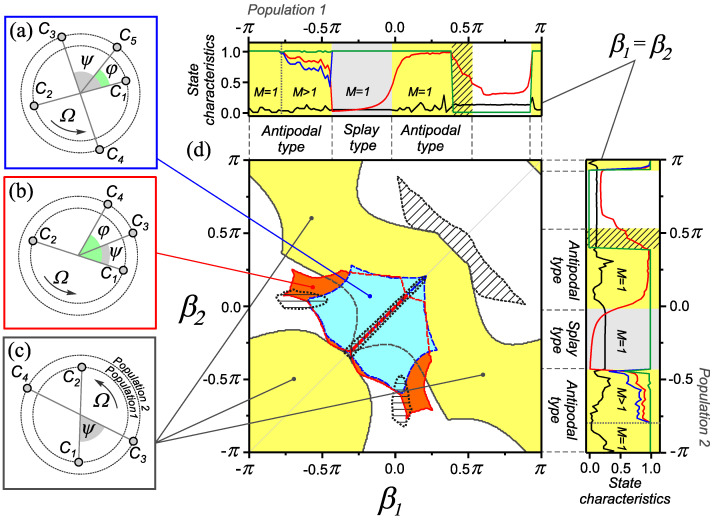
Examples of one-frequency generalized cluster states for the adaptive network (Equation 1) and (Equation 2) and regions of their existence on the parameter plane $\left(\beta_{1},\beta_{2}\right)$ . (**a**–**c**) Phase distributions for generalized cluster states similar to those shown in Figure 3. (**d**) Diagram of the dynamical states in the parameter plane $\left(\beta_{1},\beta_{2}\right)$  for $\varepsilon_{11}=\varepsilon_{22}=0.01$ , $\varepsilon_{12}=\varepsilon_{21}=0.1$  and α=0.32π . The dependencies of the characteristics (Equation 3)–(Equation 6) for individual populations shown in Figure 5a are depicted along the edges of the two-parameter diagram.

**Figure 7 entropy-25-00913-f007:**
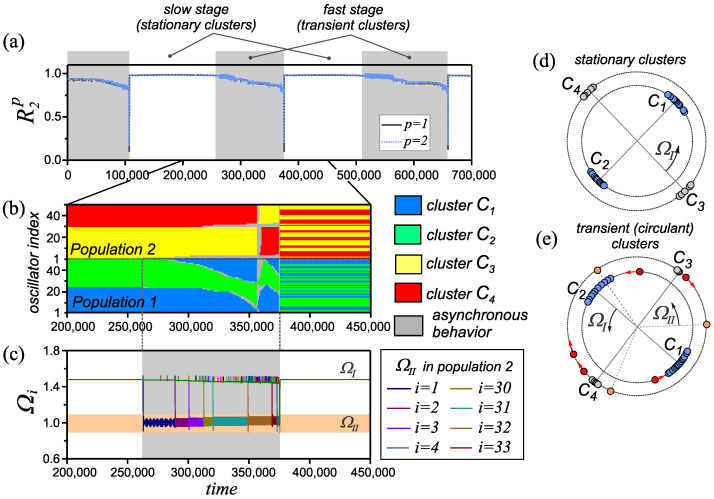
Properties of a transient cluster state in the adaptive network (Equation 1) and (Equation 2) for parameter values $\beta_{1}=0.41\pi$ , $\beta_{2}=0.35\pi$ , $\varepsilon_{11}=\varepsilon_{22}=0.01$  and $\varepsilon_{12}=\varepsilon_{21}=0.1$ . (**a**) The evolution of the order parameters $R_{2}^{p}\left(t\right)$ , p=1,2 . (**b**) Spatiotemporal diagram illustrating the restructuring of the transient phase clusters. (**c**) Evolution of instantaneous frequencies $\Omega_{i}\left(t\right)$  of network oscillators (i=1,⋯,N) . (**d**,**e**) Examples of the distribution of oscillator phases at various stages of network evolution. The symbols $\Omega_{I}$  and $\Omega_{II}$  denote the average frequencies of the corresponding frequency groups of oscillators that exist in the network at various stages of its evolution.

**Table 1 entropy-25-00913-t001:** The classification of the dynamical states of the adaptive network in accordance with the characteristics (Equation 3)–(Equation 6).

Network State	$\langle R_{1}\rangle$	$\langle R_{2}\rangle$	$R_{\mathrm{link}}$	$P_{C}$
Global network synchronization	1	1	1	1
One synchronous group (antipodal type)	∼0	1	1	1
One synchronous group (splay type)	∼0	∼0	1	1
*M* synchronous group with different frequency (antipodal type)	∼0	(0,1)	(0,1)	1
*M* synchronous group with different frequency (splay type)	∼0	∼0	(0,1)	1
Chimera state	∼0	(0,1) ^1^	(0,1)	(0,1)
Asynchronous behavior	0	0	0	0

^1^ The value of the parameter depends on the type of behavior (antipodal or splay) of the coherent part of the network.

## Data Availability

Data are available upon request from the corresponding author.

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
