# Peer review of "Transient Phase Clusters in a Two-Population Network of Kuramoto Oscillators with Heterogeneous Adaptive Interaction"

_entropy, 2023, doi:10.3390/e25060913_

Round 1
Reviewer 1 Report
The manuscript investigates a system of Kuramoto-Sakaguchi phase oscillators organized in two populations coupled with an additional dynamical adaptation (coevolution) of the coupling weights. An important novelty in the paper is the heterogeneity of the adaptive couplings. In particular, network interactions can be characterized by different time scales of coupling changes and different adaptation functions.
The authors report on new collective phenomena arising from various schemes of heterogeneous adaptation. For example, the introduction of heterogeneity in the type of adaptation leads to the suppression of splay states and the appearance of many cluster states. The presence of multistability is the reason for the complex non-stationary behavior called circulant clusters. Another type of non-stationary behavior in the form of alternating asynchronous and synchronous network dynamics associated with the emergence of antipodal clusters occurs when there is a heterogeneity in the rate of coupling adaptation. Such dynamics was not observed in the case of homogeneous adaptation.
In my opinion, the manuscript presents an interesting contribution to the relatively new field of adaptive networks and therefore deserves publication in Entropy. I recommend it for publication after considering the following comments:
- Please explain where the system of equations for \psi and \varphi comes from on page 7.
- Explain the symbols \Omega_I and \Omega_II in figures 6(d),(e).
Minor formatting issues:
- Correct figure reference on page 6, line 175. Apparently, "Figure 3(a)" should be replaced by "Figure 3(c)".
- Correct "transie behavior" on page 10, line 310.
Author Response
Response to Reviewer 1 Comments
We thank the Reviewer for a careful reading of our work and valuable comments.
In the revised version, we have rewritten and structured Section 2.2 and Section 3.1. We have made minor changes to Figures 2, 3 and 7 (Figure 6 in the original version). We have also added a new Figure (Figure 4 in the revised version). In the course of correcting Section 2.2, we also added new references to the literature. The main changes are marked in blue in the attached pdf-file of the revised manuscript (manuscript-rev(marked).pdf)
Point 1: Please explain where the system of equations for \psi and \varphi comes from on page 7.
Response 1: We have rewritten Section 3.1. In the revised version, Subsection 3.1 (on page 8) contains a detailed explanation of where the system of equations for \psi and \varphi came from.
Point 2: Explain the symbols \Omega_I and \Omega_II in figures 6(d),(e).
Response 2: \Omega_I and \Omega_II in Figure 6(d),(e) (in the revised version it is Figure 7) there is one frequency group and all network oscillators are characterized by the average frequency \Omega_I. Figure 6(e) shows the instantaneous phase distribution for the fast stage of network evolution. In this case, the second frequency group is formed, the oscillators of which are characterized by the average frequency \Omega_II. Appropriate explanations are added to the description of Figure 6.
Point 3: Correct figure reference on page 6, line 175. Apparently, "Figure 3(a)" should be replaced by "Figure 3(c)".
Response 3: We have rewritten Section 3.1. Now the description and reference to the figure are correct.
Point 4: Correct "transie behavior" on page 10, line 310.
Response 4: It is corrected. We have replaced "transie behavior" with "transient behavior".
Reviewer 2 Report
The authors analyze different synchronization regimes in a network of two populations of phase oscillators with heterogeneous adaptive mechanisms by a combination of (mostly) numerical simulation and (some) analytical reasoning. This is an interesting contribution to the study of synchronization in adaptive networks of oscillators. In a topic where it is relatively easy to focus on very specific models and parameter choices, and find novel numerical results whose generality is hard to assess, I appreciate the fact that the authors have decided instead to propose a relatively general model and a systematic study of the dynamical regimes based on the parameter variations, in order to shed some light on the problem at hand. Having said this, and despite my overall positive impression, I believe that a thorough revision of parts of the text, one that is especially focused on clarifying the explanation of some results, should be undertaken before the manuscript can be considered acceptable for publication.
Main issues:
1. In lines 88 and 89, I believe that the superindex in \phi_i^l should be 1, not l. Moreover, the \Lambda-notation of the adaptation function in those lines and Fig. 1 is not consistent with that used in line 74. In general the choice of letters for indices is a bit strange: why are the populations l and q, and the number of oscillators in populations is N_p? In fact, p is closer in the alphabet to q… moreover, does p play any role at all (could N_p be just written as N/2)? Perhaps the two populations could be denoted p and q instead of l and q... The authors should carefully revise their notation and make sure it is consistent and clear.
2. In subsection 2.2 the authors should provide some indications as to when the different observables are expected to give results close to 1, to 0 and somewhere in between, instead of just leaving reader to work out this connections by himself/herself from the dynamic regimes shown in table that is provided. On the other hand, is R_{ij} = 1 only when the phase difference between those oscillators is exactly a constant? For finite-size networks one expects the existence of certain fluctuations…, so maybe the authors consider the situation where for long times the phase difference is bounded. In Eq. (4), unless the “number of connections” N_L is twice the number of links in the network, it appears that there is a factor of ½ missing, because there may be 2*N_L terms contributing to the sum. The authors should revise this section and provide a more accurate description of the dynamical regimes underlying different possible values of the observables under consideration.
3. Even for the homogeneous case, it seems that global network synchronization (R_1 = 1) cannot be achieved for any value of beta. Do the authors have an explanation for this? Is this expected to change for other values of \alpha (such as \alpha = 0)?
4. The paragraph about “one-frequency cluster states” at the very end of Section 2 is very unclear. The authors should be more specific about those states (which later on play an important role in the discussion), why they require specific initial conditions within small basins of attractions, etc.
5. Subsection 3.1 is unnecessarily hard to read. It should be revised so as to make it clearer in at least in two ways: what is written there should be certainly improved, and more explanations should be provided at each step. In short, the discussion should be more accessible to the reader. Some points are indeed very confusing: 1) Isn’t the comment in line 175 referring to Fig. 3 (c)? 2) In Eq. (8) where has Eq. (7) been used? In general the connection between the equations is unnecessarily hard at the moment. 3) Can Eq. (10) be somewhat simplified? Or, at the very least, can the conditions for its derivation be described in words (so that someone who wants to use it can recover it with some symbolic manipulation software such as Mathematica)? The discussion immediately before (10) is especially hard to follow.
6. The circulant clusters described at the end of Subsection 3.1 are very intriguing, and they certainly deserve some graphical illustration based on numerical data, as that provided in Fig. 4 for transient pulsating clusters (but shorter, as they have been described elsewhere).
7. While the analyses of results contained in Subsections 3.2 and 3.3 are much clearer than those of Subsection 3.1, they would also benefit from some revision.
Minor points:
8. In the introduction references dealing with “models of adaptively coupled phase oscillators” are cited, but the list is far from complete. For example, publications such as Phys. Rev. Lett. 107, 234103 (2011), Sci. Rep. 1, 99 (2011) and Phys. Rev. E 86, 015101(R) (2012), showing the emergence of structural features out of adaptive mechanisms in networks of phase oscillators, are not cited, and I doubt that these are the only influential publications that have been left out.
9. At the end of page 2 connections to adaptive mechanisms studied in the context of biological neural networks, such as spike-timing-dependent plasticity and Hebbian learning, are mentioned quite briefly. As neuroscience provides the main motivation for many studies of adaptive synchronization, features prominently in the concluding paragraph of the manuscript, and not all readers with a background in physics or applied mathematics are familiar with it, I think that a short paragraph describing this connection in greater detail is required.
10. Typos: 1) A comma should be removed in lines 95 and 143; 2) in the title of subsection 2.2 ‘for’ appears twice; 3) “sections3.1” (without space) in line 300; 3) “transie” in line 310.
Author Response
Response to Reviewer 2 Comments
We thank the Reviewer for a careful reading of our work and valuable comments.
In the revised version, we have rewritten and structured Section 2.2 and Section 3.1. We have made minor changes to Figures 2, 3 and 7 (Figure 6 in the original version). We have also added a new Figure (Figure 4 in the revised version). In the course of correcting Section 2.2, we also added new references to the literature. The main changes are marked in blue in the attached pdf-file of the revised manuscript (manuscript-rev(marked).pdf)
Point 1: 1. In lines 88 and 89, I believe that the superindex in \phi_i^l should be 1, not l. Moreover, the \Lambda-notation of the adaptation function in those lines and Fig. 1 is not consistent with that used in line 74. In general the choice of letters for indices is a bit strange: why are the populations l and q, and the number of oscillators in populations is N_p? In fact, p is closer in the alphabet to q… moreover, does p play any role at all (could N_p be just written as N/2)? Perhaps the two populations could be denoted p and q instead of l and q... The authors should carefully revise their notation and make sure it is consistent and clear.
Response 1: We have changed the index designations in accordance with the recommendations of the reviewer. We also abandoned the notation for the number of oscillators in populations N_p. In the text of the article, we have replaced "N_p" with "N/2".
Point 2: In subsection 2.2 the authors should provide some indications as to when the different observables are expected to give results close to 1, to 0 and somewhere in between, instead of just leaving reader to work out this connections by himself/herself from the dynamic regimes shown in table that is provided. On the other hand, is R_{ij} = 1 only when the phase difference between those oscillators is exactly a constant? For finite-size networks one expects the existence of certain fluctuations…, so maybe the authors consider the situation where for long times the phase difference is bounded. In Eq. (4), unless the “number of connections” N_L is twice the number of links in the network, it appears that there is a factor of ½ missing, because there may be 2*N_L terms contributing to the sum. The authors should revise this section and provide a more accurate description of the dynamical regimes underlying different possible values of the observables under consideration..
Response 2: We have described in more detail the relationship between the characteristics and modes observed in the network. We also described the procedure for calculating the mutual synchronization parameters R_{ij} and explained their use for the introduced characteristics. When calculating R_{link} in equation (4), we really considered N_L to be twice the number of links. In the revised version of the manuscript, we have corrected this inaccuracy.
Point 3: Even for the homogeneous case, it seems that global network synchronization (R_1 = 1) cannot be achieved for any value of beta. Do the authors have an explanation for this? Is this expected to change for other values of \alpha (such as \alpha = 0)?
Response 3: In fact, the global synchronization mode is a special case of a synchronous state of the antipodal type, when the size of one of the phase clusters is equal to 0. Accordingly, the domains of existence of these modes in the parameter space also coincide. However, the global synchronization mode is observed when the initial conditions are appropriately specified. In particular, this mode can be obtained by selecting \kappa_{ij}(0) in the range of positive values. Figure 2 is constructed for a more general case, when the initial values $\kappa_{ij}$ are randomly distributed in the range $[-1,1]$. Therefore, the global synchronization mode is not explicitly indicated in Figure 2. The corresponding comment about the global synchronization mode is included in Subsection 2.2 (on page 5).
Point 4: The paragraph about “one-frequency cluster states” at the very end of Section 2 is very unclear. The authors should be more specific about those states (which later on play an important role in the discussion), why they require specific initial conditions within small basins of attractions, etc.
Response 4: We have rewritten the paragraph about one-frequency cluster states in Subsection 2.2 (on page 6).
Point 5: Subsection 3.1 is unnecessarily hard to read. It should be revised so as to make it clearer in at least in two ways: what is written there should be certainly improved, and more explanations should be provided at each step. In short, the discussion should be more accessible to the reader. Some points are indeed very confusing: 1) Isn’t the comment in line 175 referring to Fig. 3 (c)? 2) In Eq. (8) where has Eq. (7) been used? In general the connection between the equations is unnecessarily hard at the moment. 3) Can Eq. (10) be somewhat simplified? Or, at the very least, can the conditions for its derivation be described in words (so that someone who wants to use it can recover it with some symbolic manipulation software such as Mathematica)? The discussion immediately before (10) is especially hard to follow.
Response 5: We have rewritten Subsection 3.1. The results presented in Subsection 3.1. were structured. References to figures and descriptions in the text have been corrected. We have added a detailed description of the derivation of equation (10) (in the revised version, equation (11)).
Point 6: The circulant clusters described at the end of Subsection 3.1 are very intriguing, and they certainly deserve some graphical illustration based on numerical data, as that provided in Fig. 4 for transient pulsating clusters (but shorter, as they have been described elsewhere).
Response 6: We have added a graphical illustration (Figure 4 in revision version) of the state of circulant clusters in Subsection 3.1.
Point 7: While the analyses of results contained in Subsections 3.2 and 3.3 are much clearer than those of Subsection 3.1, they would also benefit from some revision.
Response 7: We have made some revision of Subsections 3.2 and 3.3.
Point 8: In the introduction references dealing with “models of adaptively coupled phase oscillators” are cited, but the list is far from complete. For example, publications such as Phys. Rev. Lett. 107, 234103 (2011), Sci. Rep. 1, 99 (2011) and Phys. Rev. E 86, 015101(R) (2012), showing the emergence of structural features out of adaptive mechanisms in networks of phase oscillators, are not cited, and I doubt that these are the only influential publications that have been left out.
Response 8: We have added references to these articles when discussing models of adaptively coupled phase oscillators in the Introduction on page 1.
Point 9: At the end of page 2 connections to adaptive mechanisms studied in the context of biological neural networks, such as spike-timing-dependent plasticity and Hebbian learning, are mentioned quite briefly. As neuroscience provides the main motivation for many studies of adaptive synchronization, features prominently in the concluding paragraph of the manuscript, and not all readers with a background in physics or applied mathematics are familiar with it, I think that a short paragraph describing this connection in greater detail is required.
Response 9: We have rewritten a paragraph (on pages 2-3) describing the relationship between the form of the adaptation function and the mechanisms of plasticity that can be found in neural networks.
Point 10: Typos: 1) A comma should be removed in lines 95 and 143; 2) in the title of subsection 2.2 ‘for’ appears twice; 3) “sections3.1” (without space) in line 300; 3) “transie” in line 310.
Response 10: All typos corrected.
Round 2
Reviewer 1 Report
The authors responded to all of my comments and added the corresponding changes to the manuscript. I recommend publishing the paper in Entropy.
Reviewer 2 Report
The authors have satisfactorily addressed the comments of my previous report. The manuscript has improved as a result of these and other modifications, and I recommend its publication in its present form.